# Modelling the number of antenatal care visits in Bangladesh to determine the risk factors for reduced antenatal care attendance

Kakoli Rani Bhowmik[1], Sumonkanti Das[2,3,4], Md. Atiqul Islam[3] *

1 Department of Medical Decision Making, Leiden University Medical Center, Leiden, Netherlands,
2 Department of Quantitative Economics, Maastricht University, Maastricht, Netherlands, 3 Department of Statistics, Shahjalal University of Science & Technology, Sylhet, Bangladesh, 4 Faculty of Engineering and Information Sciences, University of Wollongong, Wollongong, NSW, Australia

☯ These authors contributed equally to this work.
* atique-sta@sust.edu

**Data Availability Statement:** All relevant data are within the manuscript and its Supporting Information files. The additional data is also available from DHS (https://www.dhsprogram.

## Abstract

The existence of excess zeros in the distribution of antenatal care (ANC) visits in Bangladesh raises the research question of whether there are two separate generating processes in taking ANC and the frequency of ANC. Thus the main objective of this study is to identify a proper count regression model for the number of ANC visits by pregnant women in Bangladesh covering the issues of overdispersion, zero-inflation, and intra-cluster correlation with an additional objective of determining risk factors for ANC use and its frequency. The data have been extracted from the nationally representative 2014 Bangladesh Demographic and Health Survey, where 22% of the total 4493 women did not take any ANC during pregnancy. Since these zero ANC visits can be either structural or sampling zeros, two-part zero-inflated and hurdle regression models are investigated along with the standard one-part count regression models. Correlation among response values has been accounted for by incorporating cluster-specific random effects in the models. The hurdle negative binomial regression model with cluster-specific random intercepts in both the zero and the count part is found to be the best model according to various diagnostic tools including likelihood ratio and uniformity tests. The results show that women who have poor education, live in poor households, have less access to mass media, or belong to the Sylhet and Chittagong regions are less likely to use ANC and also have fewer ANC visits. Additionally, women who live in rural areas, depend on family members' decisions to take health care, and have unintended pregnancies had fewer ANC visits. The findings recommend taking both cluster-specific random effects and overdispersion and zero-inflation into account in modelling the ANC data of Bangladesh. Moreover, safe motherhood programmes still need to pay particular attention to disadvantaged and vulnerable subgroups of women.

com) upon request. The authors did not receive special access privileges to the data that others would not have.

**Funding:** The author(s) received no specific funding for this work.

**Competing interests:** The authors have declared that no competing interests exist.

# Introduction

Following the third Sustainable Development Goal of reducing the global maternal mortality ratio (MMR) to 70 per 100,000 live births by 2030 [1, 2], the Bangladesh government set targets to reduce the MMR to 143 and 105 per 100,000 live births, in 2015 and 2022 respectively. Though Bangladesh has significantly improved the MMR, the improvement stagnated at 196 per 100,000 live births, according to the Bangladesh Maternal Mortality and Health Care Surveys conducted in 2010 and 2016 [3]. Access to maternal health service for all women during their pregnancy period and childbirth is crucial to further reduce pregnancy-related morbidity and mortality. According to the WHO [4], antenatal care visits (ANC) should include at least four visits to medically trained personnel to avoid complications and have a safe delivery. Despite relatively high rates of ANC utilization and improved access to health facilities in Bangladesh, the number of pregnant women with at least this number of four ANC visits remains low (at 31%) and the number of women without any ANC visits remains high (at 22%) [5].

Many studies have assessed the determinants of prenatal care attendance in Bangladesh, particularly focussing on the WHO guideline of taking at least four ANC visits [6–9], but only a few studies have assessed the determinants of the number of ANC visits in general [10–13]. It has been shown that there may be two separate processes that generate decisions regarding the use and frequency of prenatal care use [14, 15]. Without distinguishing these generating processes, Poisson regression (PR) and negative binomial regression (NBR) models have been widely used to model the number of ANC visits in Bangladesh [12, 13, 16]. But these models may provide inconsistent regression coefficient, as overdispersion and excess zeros remain unaccounted for [17]. In such situations, zero-inflated and hurdle regression (ZIR and HR) models can be applied [18]. These so-called two-part models have been used in a limited number of studies [10, 11]. In addition to the problems of overdispersion and zero-inflation in the ANC data, correlation among measurements (a common phenomenon in longitudinal, repeated surveys, and clustered data) known as intra-cluster correlation (ICC) needs to be considered [19]. The problems associated with ICC can be partially solved by incorporating cluster specific random effects in the standard PR, NBR, HR, and ZIR models [20, 21]. Guliani, Sepehri and Serieux [22] employed a two-part HR model incorporating such cluster effects in the model and explored the determinants of ANC use and the frequency of ANC visits utilizing ANC data from 32 developing countries.

To the best of our knowledge, no study has simultaneously considered all the issues discussed above in modelling the number of ANC visits of Bangladeshi women. Thus, the objectives of this study are two-fold: (i) to develop a proper count regression model for the number of ANC visits in Bangladesh covering the issues of overdispersion, zero-inflation, and ICC; and (ii) to determine the risk factors for no ANC use as well as the frequency of ANC visits.

# Methods

## Data description

In this study, data are extracted from the nationally representative 2014 Bangladesh Demographic and Health Survey (BDHS), where the country was stratified into 20 sampling strata according to urban and rural enumeration areas of 7 divisions [5]. A two-stage stratified sampling design was implemented to collect data: in the first stage 600 clusters (393 from rural and 207 from urban areas) were drawn with probability proportional to the enumeration area size and in the second stage 30 households per cluster were selected with an equal probability systematic procedure. The 2014 BDHS covers 17,863 ever-married women aged 15–49 years from 17,300 households. The information on ANC visits was collected from 4493 ever

married-women who gave birth in the three years preceding the survey. Among women with two or more live births within the given period, information was only recorded for the last birth. Mothers were asked a number of questions about ANC visits and the received health care during the antenatal visits. The number of ANC visits (a non-negative integer) is the target response variable for which this study aims to identify a proper count regression model. A number of explanatory variables at individual (woman), household, community, and regional levels have been considered based on recent studies on ANC utilization [6, 9, 13]. The bivariate relationship of the explanatory variables with the number of ANC visits were examined first by a developing simple PR model for each of the explanatory variables. Individual-level explanatory variables in this study include education status of the women and their husbands, women's access to mass media exposure, women's decision-making power on their own healthcare issues, and women's desire of pregnancy along with household wealth status, and place of residence, and regional settings.

## Statistical models

Let $y_{ij}$ denote the number of ANC visits of the $i^{th}$ women living in the $j^{th}$ cluster, and the vector $X_{ij}$ the corresponding values of the considered explanatory variables. Assuming independence of ANC visits of the women, the PR and NBR models are defined by:

$$log(\mu_{ij}) = \beta_0 + \boldsymbol{\beta}X_{ij}^T$$

where $\mu_{ij}$ is the expected number of ANC visits as a function of explanatory variables, $\beta_0$ the overall intercept, and $\boldsymbol{\beta}$ the vector of regression coefficients. The difference between the PR and the NBR model is in the assumed distribution of $y_{ij}$ in the respective models. In PR, the response variable is assumed to follow a Poisson distribution with $E(y_{ij}) = \mu_{ij} = var(y_{ij})$, while in NBR it is assumed to follow a negative binomial distribution with $E(y_{ij}) = \mu_{ij}$ and $var(y_{ij}) = \mu_{ij} + \mu_{ij}^2/\theta$, where $\theta$ is the shape parameter which controls the dispersion. When $\theta \to \infty$, the NBR model converges to a PR model without overdispersion. Thus, the PR model is a limiting model of the NBR model as the dispersion $(\mu_{ij}^2/\theta)$ approaches zero.

The PR and NBR models assume that the observations conditioned on the predictors are independent and identically distributed [23]. However, these assumptions may be violated in clustered data. Ignoring possible correlation in the data result in a model that could lead to biased estimates and misinterpretation of the results [24]. An acceptable way of accommodating this non-independence of observations is to use mixed-effects models, also known as multilevel models. The use of a multilevel modelling strategy accommodates the clustered or hierarchical nature of the BDHS data and corrects standard errors of the estimated coefficients for ICC. A simple mixed-effects PR/NBR model is obtained by incorporating cluster-specific random effects in the standard PR/NBR model:

$$log(\mu_{ij}) = (\beta_0 + b_{oj}) + \boldsymbol{\beta}X_{ij}^T$$

where $b_{oj}$ stand for the random intercepts at cluster level and are assumed to follow a normal distribution with constant variance. The mixed-effects PR and NBR models are referred to hereafter as MPR and MNBR models respectively.

The zero-inflated and hurdle extensions of the PR and NBR models are the most prominent and effective models not only to handle excess zeros in a count data but also to accommodate overdispersion resulting from the variance being greater than the mean [18]. Both ZIR and HR models have a mixture of two generating processes. In the ZIR model, the first process generates only zero counts (structural or genuine zeros) with probability $\varphi_{ij}$, while the second

process generates non-negative counts (which could result in zeros called sampling zeros) from either a Poisson or a negative binomial distribution with probability $(1-\varphi_{ij})$. Like the ZIR model, the HR models also assume that the first process generates only structural or genuine zeroes while the second process generates truncated positive counts from a zero-truncated Poisson or negative binomial distribution [25]. In relation to ANC visits, structural zeros occur if a pregnant woman would never visit and sampling zeros occur if she could visit but has no reason to do so within the specified time frame. In this study, women reported the ANC visit for their last birth over the three years preceding the survey. Accordingly, the reported zeros could be structural, but they could also be sampling errors due to incorrect recall or misspecification of the time frame. In the ZIR model, the distribution of the number of ANC visits is modeled as:

$$P[y_{ij} = 0] = \varphi_{ij} + (1 - \varphi_{ij})f_{ij}(0) \text{ and}$$

$$P[y_{ij} = y] = (1 - \varphi_{ij})f_{ij}(y) \text{ for } y \geq 1$$

where $f_{ij}(.)$is either a Poisson or a negative binomial distribution.

The basic difference between the two models is that the ZIR model uses two distribution for the zero counts, while the HR model use one distribution for the zero counts:

$$P[y_{ij} = 0] = \varphi_{ij} \text{ and}$$

$$P\left[y_{ij} = y\right] = \left(1 - \varphi_{ij}\right)\frac{f_{ij}(y)}{1 - f_{ij}(0)} \text{ for } y \geq 1$$

The first so-called zero part of the process can be modelled as a binary or logit model. According to the modelling distribution (Poisson or negative binomial) of the second so-called count part of the process, the ZIR and HR models are referred to as either ZIPR/HPR or ZINBR/HNBR.

In general, separate explanatory variables can be used in the two parts of the ZIR and HR models. To explain these models mathematically, let $X_{ij}$ and $Z_{ij}$ be vectors of known explanatory variables used in the zero- and count-part models respectively. Then, the zero and count part models under the simple ZIR and HR models can be expressed, respectively, as:

$$logit(\varphi_{ij}) = \gamma_0 + \gamma Z_{ij}^T \quad \text{and} \quad log(\mu_{ij}) = \beta_0 + \beta X_{ij}^T$$

where $\gamma_0$ is the overall intercept, and $\gamma$ is the vector of regression coefficients of the binary process (binary logistic model). The HR model separates the structural zeros from the non-zero responses by modelling non-zero counts with a truncated Poisson/negative binomial distribution. Consequently, the effects of covariates on $\varphi_{ij}$ in the HR model (on the log-odds of a structural zero) and their effects on $\varphi_{ij}$ in ZIR model (on the log-odds of a structural and sampling zero) are not equivalent [26, 27]. The mixture of two parts in a ZIR/HR model allows to interpret separate answers of the two questions (i) which factors influence whether a pregnant woman will attend ANC or not and (ii) which factors predict the number of times she will take ANC. Moreover, explanatory variables may have different impacts in the two processes.

The mixed-effects ZIR and HR models can be expressed by adding a cluster-specific random component $b_{0j}$ to $\beta_0$ in the count part and another cluster-specific random component $c_{0j}$ to $\gamma_0$ in the zero part:

$$logit(\varphi_{ij}) = (\gamma_0 + c_{0j}) + \gamma Z_{ij}^T \text{ and } log(\mu_{ij}) = (\beta_0 + b_{0j}) + \beta X_{ij}^T.$$

When cluster-specific random effects are considered only in the count part (so $c_{0j} = 0$), the mixed-effects ZIR and HR models are denoted by MZIPR/MZINBR and MHPR/MHNBR respectively depending on the assumed distribution (Poisson or negative binomial) of the count part. Models with also a random effect in the zero part (usually it is considered as an extra random effect in the mixed-effects ZIR/HR model) are denoted hereafter as MZINBR. ERE (for example) for the MZINBR model.

The two model processes or their log-likelihoods are assumed functionally independent, so the joint likelihood can be maximized by maximizing each part separately [26]. A maximum likelihood method approximating the integrals over the random effects with an adaptive Gaussian quadrature rule [28] was used to fit the mixed-effects ZIR and HR models. Several R-packages were used to analyse different versions of the PR and NBR models. The recently developed "GLMMadaptive" package of Rizopoulos [29] was employed to fit mixed-effects ZIR and HR models. Significance of the dispersion parameter, zero-inflation, and goodness-of-fit of the model ($H_0$: fitted model suits well for the data) which is further referred to as the uniformity test were assessed using the residual diagnostics for hierarchical (multi-level/mixed) regression models available in the DHARMa (Diagnostics for Hierarchical Regression Models) package of Hartig [30].

Since the considered models are based on different assumptions, their direct comparison is complicated. A step by step comparison procedure is followed providing priority to the uniformity test and significance of the cluster-specific random effects to select the final model. The basic steps are:

**Step 1:** First PR and NBR models with cluster-specific random intercept (MPR and MNBR) are examined assessing whether overdispersion and zero-inflation issues are covered by the fitted models.

**Step 2:** If zero-inflation remains, ZIR and HR models with (MZIPR, MZINBR, MHPR, MHNBR) or without (ZIPR, ZINBR, HPR, HNBR) cluster-specific random intercept are estimated and compared. Mixed-effects ZIR and HR models are developed considering cluster-specific random intercept only in the count part (say, MZINBR) as well as in both the count and the zero parts (say, MZINBR.ERE). Nested and non-nested models are compared using likelihood ratio (LR) and Vuong tests [31] respectively.

**Step 3:** The final model is selected using the DHARMa's uniformity test, assessing which model fits better for the data. Since mixed-effects ZIR and HR models are not directly comparable, the uniformity test is used to examine whether MZINBR or MHNBR (with or without extra cluster-specific random effects) suits the studied data better. Significance of the cluster variance component in the zero and count parts is assessed using the LR test.

## Results

The distribution of the number of ANC visits shown in Fig 1 is positively skewed with low mean (2.75) and median (2.0) number of ANC visits. About 22% of the pregnant women did not take any ANC visits and only 31% took ANC at least 4 times during their pregnancy period. Table 1 shows mean and median numbers of ANC visits according to different background characteristics of the women. Both the mean and the median frequency significantly vary with all these characteristics. Women from the Khulna division had a higher mean (3.42) and median (3) number of ANC visits and those from the Sylhet division (2.02 and 1 respectively) had the lowest. As expected, urban women have a higher mean and median number of ANC visits than the rural women. Women's education and exposure to mass media (TV),

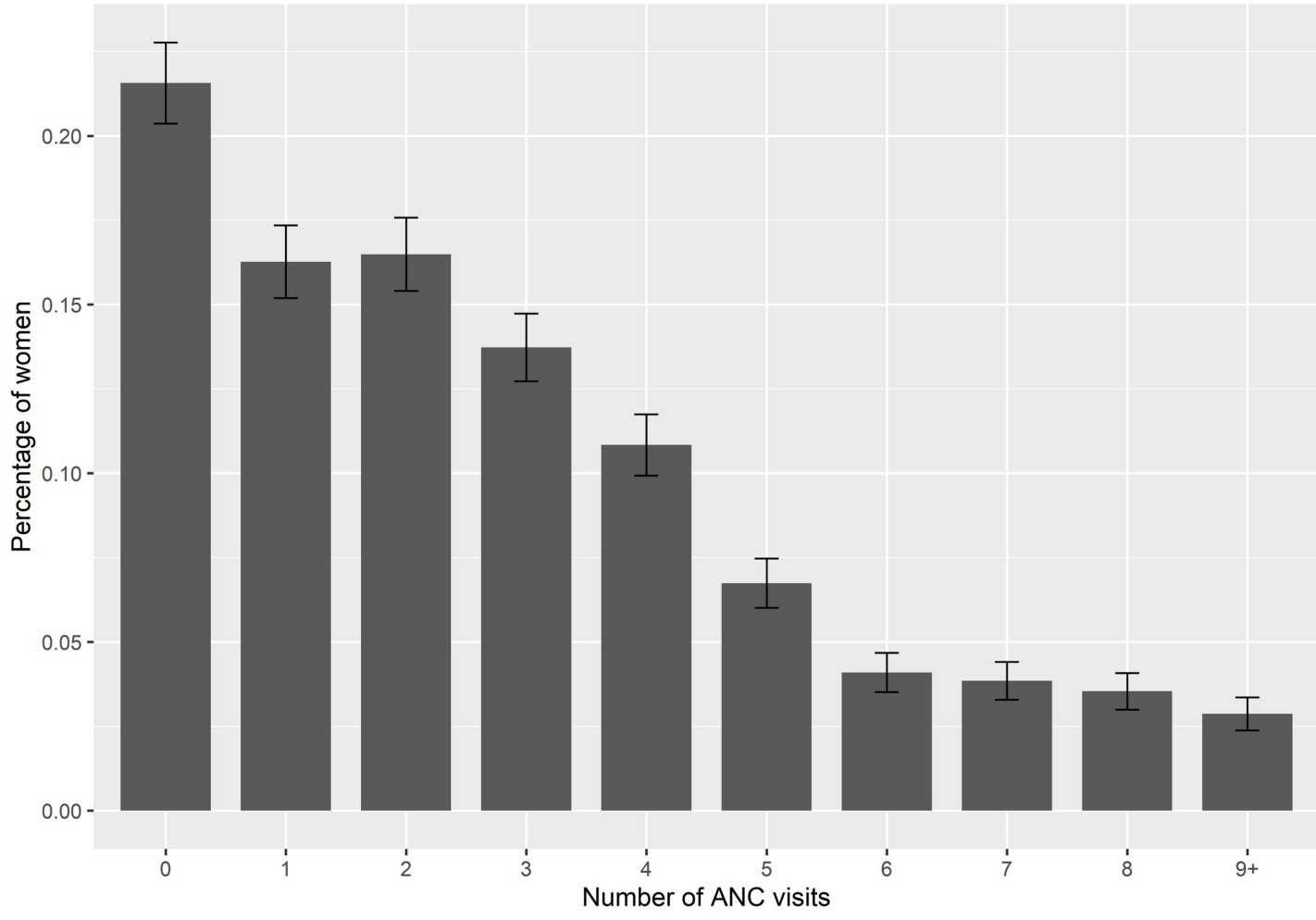

**Fig 1. Distribution of the number of antenatal care (ANC) visits (with 95% confidence interval) among pregnant women in Bangladesh.**

their husbands' education and the household wealth status showed a significant positive association with the mean and median number of ANC visits. Women who wanted their pregnancy had a higher mean and median number of ANC visits than those with an unwanted pregnancy. Women's decision-making power on their own healthcare issues showed significant association with the mean and median number of ANC visits.

## Model selection

Policy makers, stakeholders, and donors explore risk factors for reduced prenatal care use or lower frequency of visits to design strategies to improve maternal health care. A proper count regression model for the number of ANC visits incorporating multiple factors helps to identify those core risk factors. In this study, one-part regression models (such as PR and NBR) and two-part regression models (such as ZIR or HR) with and without consideration of ICC are compared to examine whether there are indeed two generating processes in the number of ANC visits as well as to determine the risk factors of the processes.

A fixed set of explanatory variables was used in all models of ANC visits for comparison purposes. The comparison of the standard PR and NBR models with their mixed-effects models MPR and MNBR shown in Table 2 indicates that PR and MPR models fail to capture the

**Table 1. Mean and median number of antenatal care (ANC) visits in Bangladesh according to women's background characteristics, and the associated statistical F-test and Chi-Square ($\chi^2$) test for equality of the means and medians, respectively.**

| Factors | Category | Sample Size | Mean | F (p-value) | Median | $\chi^2$ (p-value) |
|---|---|---|---|---|---|---|
| Region | Barisal | 532 | 2.43 | 22.62 (0.000) | 2 | 168.86 (0.000) |
| | Chittagong | 862 | 2.57 | | 2 | |
| | Dhaka | 795 | 3.04 | | 2 | |
| | Khulna | 531 | 3.42 | | 3 | |
| | Rajshahi | 546 | 2.82 | | 2 | |
| | Rangpur | 549 | 3.20 | | 3 | |
| | Sylhet | 578 | 2.02 | | 1 | |
| Place of Residence | Urban | 1451 | 3.66 | 281.87 (0.000) | 3 | 261.89 (0.000) |
| | Rural | 3042 | 2.34 | | 2 | |
| Woman's Education | Illiterate | 606 | 1.50 | 215.85 (0.000) | 1 | 618.04 (0.000) |
| | Primary | 1235 | 2.03 | | 1 | |
| | Secondary | 2130 | 3.06 | | 3 | |
| | Higher | 522 | 4.69 | | 4 | |
| Wealth Status | Poorest | 939 | 1.62 | 191.31 (0.000) | 1 | 728.80 (0.000) |
| | Poorer | 855 | 2.08 | | 2 | |
| | Middle | 860 | 2.49 | | 2 | |
| | Richer | 946 | 3.23 | | 3 | |
| | Richest | 893 | 4.39 | | 4 | |
| Mass Media Exposure | Not at all | 1852 | 1.86 | 246.75 (0.000) | 1 | 533.34 (0.000) |
| | Less than once a week | 396 | 2.57 | | 2 | |
| | At least once a week | 2245 | 3.55 | | 3 | |
| Pregnancy wanted | No | 462 | 1.92 | 57.48 (0.000) | 1 | 68.59 (0.000) |
| | Yes | 4031 | 2.86 | | 2 | |
| Decision on health care | Woman alone | 529 | 3.07 | 9.87 (0.000) | 3 | 34.83 (0.000) |
| | Woman & husband | 2145 | 2.89 | | 2 | |
| | Husband alone | 1429 | 2.49 | | 2 | |
| | Other | 390 | 2.67 | | 2 | |
| Partner's Education | Illiterate | 1030 | 1.82 | 183.56 (0.000) | 1 | 541.60 (0.000) |
| | Primary | 1353 | 2.29 | | 2 | |
| | Secondary | 1420 | 3.13 | | 3 | |
| | Higher | 690 | 4.37 | | 4 | |
| National | | 4493 | 2.76 | | 2 | |

overdispersion, while both NBR and MNBR do account for overdispersion, but all four models are unable to account for the issue of zero-inflation. Among these models, the NBR model is preferred by the DHARMa uniformity test with lower p-value (0.066). However the AIC, log-

**Table 2. Akaike's information criteria (AIC), log-likelihood, likelihood-ratio (LR), dispersion, zero-inflation and uniformity tests (based on Kolmogorov's D-statistic) for the one-part Poisson regression (PR), negative-binomial regression (NBR), mixed PR (MPR) and Mixed NBR (MNBR) models.**

| Model | AIC | log-likelihood (df) | LR test (p-value) | Dispersion Test (Ratio Statistic & p-value) | Zero-Inflation Test (Ratio Statistic & p-value) | Uniformity Test (D-Statistic & p-value) |
|---|---|---|---|---|---|---|
| PR | 19274.66 | -9613.33 (24) | 712.52 (0.000) | 1.358 & 0.00 | 2.001 & 0.00 | 0.114 & 0.00 |
| MPR | 18565.31 | -9257.66 (25) | | 1.096 & 0.004 | 1.685 & 0.00 | 0071 & 0.00 |
| NBR | 18314.42 | -9132.21 (25) | 216.47 (0.000) | 0.937 & 0.000 | 1.208 & 0.00 | 0.019 & 0.066 |
| MNBR | 18099.96 | -9023.98 (26) | | 0.913 & 0.002 | 1.223 & 0.00 | 0.022 & 0.026 |

**Table 3. Vuong Tests for the non-nested models Poisson regression (PR), negative-binomial regression (NBR), zero-inflated PR (ZIPR), hurdle PR (HPR), zero-inflated NBR (ZINBR), and hurdle NBR (HNBR) models.**

| Model 1 | Model 2 | Test Statistic (AIC Corrected) | p-value | Better Model |
|---------|---------|-------------------------------|---------|--------------|
| PR | NBR | -11.852 | < 2.22e-16 | NBR |
| PR | ZIPR | -12.229 | < 2.22e-16 | ZIPR |
| PR | HPR | -12.221 | < 2.22e-16 | HPR |
| ZIPR | HPR | 0.239 | 0.405 | ZIPR/HPR |
| NBR | ZINBR | -7.674 | 8.31e-15 | ZINBR |
| NBR | HNBR | -7.719 | 9.06 e-15 | HNBR |
| ZINBR | HNBR | 0.203 | 0.420 | ZINBR/HNBR |
| ZIPR | ZINBR | -7.829 | 2.46e-15 | ZINBR |
| HPR | HNBR | -7.766 | 4.05e-15 | HNBR |

likelihood, and LR test indicate that inclusion of random intercepts are required for the studied ANC data. Thus, the ZIR and HR models without and with random intercepts were developed. The results of Vuong tests for non-nested models shown in Table 3 indicate that either ZINBR or HNBR can be considered to be the better model to account for excess zeros. Table 3 also reflects that the overdispersion is captured better by the NBR-based models than the PR-based models. The results of the LR tests for nested models shown in Table 4 indicate that cluster-specific random effects should be considered in the NBR-based models. Random effects are also found important for both the count- and the zero-part models in both cases of the ZINBR (MZINBR and MZINBR.ERE) and the HNBR (MHNBR and MHNBR.ERE) models.

Since there are four possible candidates to be the best model for the ANC data, the DHAR-Ma's uniformity test was performed to find the best suitable model among these. Table 5 shows that the ZINBR model with random intercepts in the count-part (MZINBR) confirms uniformity (p-value = 0.283) with the observed count data, but the LR test in Table 4 shows that this model still requires random intercepts in the zero part (MZINBR.ERE) (p-value < 0.001). However, the MZINBR.ERE failed the uniformity test (p-value = 0.012). On the other hand, HNBR with random intercepts in the count part (MHNBR) and HNBR with random intercepts at both the count and the zero parts (MHNBR.ERE) passed the uniformity test (p-value = 0.549 and 0.118). Thus, MHNBR.ERE is considered to be the best model among the possible candidate models for the ANC data of Bangladesh. Also, the MHNBR and MHNBR.ERE models provide lower cluster-specific variance components (as well as lower ICC) than the MZINBR and MZINBR.ERE models. Do note that the same set of explanatory variables are maintained in all the one-part models and two-part models for comparison purposes. Informal diagnoses of the cluster-specific residuals through Q-Q and distribution plots of the standardized residuals shown in Fig 2 confirm that the cluster-specific residuals obtained from both the count and the zero parts are normally distributed with constant variance.

**Table 4. Likelihood Ratio (LR) tests for the nested zero-inflated NBR (ZINBR), mixed ZINBR (MZINBR), mixed ZINBR with extra random effects in the zero part (MZINBR.ERE) models, and hurdle NBR (HNBR), mixed HNBR (MHNBR), and mixed HNBR with extra random effects in the zero-part (MHNBR.ERE) models.**

| Model 1 | Model 2 | LR Test | DF | p-value |
|---------|---------|---------|-----|---------|
| ZINBR | MZINBR | 127.09 | 1 | < 2.2e-16 |
| MZINBR | MZINBR.ERE | 65.32 | 2 | < 0.0001 |
| HNBR | MHNBR | 86.59 | 1 | < 2.2e-16 |
| MHNBR | MHNBR.ERE | 106.59 | 2 | < 0.0001 |

**Table 5. DHARMa Uniformity Test for the mixed-effect zero-inflated negative binomial (MZINBR), MZINBR with extra random effects in the zero part (MZINBR.ERE), mixed-effect hurdle negative binomial (MHNBR), and MHNBR with extra random effects in the zero part (MHNBR.ERE) models (p-value < 0.05 indicates the model doesn't fit well for the count data) and the corresponding count-part ($\sigma_C^2$) and zero-part ($\sigma_Z^2$) variance components along with the intra-cluster correlation (ICC).**

| Model | MZINBR | MZINBR.ERE | MHNBR | MHNBR.ERE |
|---|---|---|---|---|
| D-Statistic | 0.015 | 0.024 | 0.011 | 0.018 |
| p-value | 0.283 | 0.012 | 0.549 | 0.118 |
| $\sigma_C^2$ | 0.074 | 0.060 | 0.064 | 0.069 |
| $\sigma_Z^2$ | - | 1.238 | - | 0.626 |
| $\rho_c = \sigma_C^2/(\sigma_C^2 + \pi^2/3)$ | 0.022 | 0.018 | 0.019 | 0.020 |
| $\rho_z = \sigma_Z^2/(\sigma_Z^2 + \pi^2/3)$ | - | 0.273 | - | 0.160 |

## Risk factors

According to the selected HR model with random effects in both the count and the zero parts (the MHNBR.ERE model), division, place of residence, household wealth, women's media exposure, women's and their partner's education status, women's decision-making power on their own healthcare issues, and desire for pregnancy have highly significant effects on either zero prenatal care use or the frequency of prenatal care use (Fig 3 and Table 6). The count-part model shows the effects of the considered factors on the frequency of ANC visits represented as incidence rate ratio (IRR), while the zero-part model shows the effects of the considered factors on the women's decision to take no ANC represented as odds ratio (OR). The estimated IRR and OR with their 95% CI are in red and blue respectively in Fig 3. Since both parts have cluster-specific random effects, the estimated parameters represent the effects of individual-, household-, regional-, and community-level characteristics on ANC attendance and the frequency of ANC visits after controlling for the unobserved community level factors. It is noted that regression coefficients of other models are not presented here, only their summary statistics were reported for comparison purposes.

The results of the finally selected count-part model shown in Fig 3 and Table 6 indicate that women living in the Khulna (IRR: 1.18, CI: 1.03–1.34) and Rangpur division (IRR: 1.24, CI: 1.09–1.41), women from the richer (IRR: 1.15, CI: 1.04–1.28) and richest (IRR: 1.30, CI: 1.16–1.46) households, women having access to mass media at least once a week (IRR: 1.16, CI: 1.08–1.24), women attending secondary (IRR: 1.21, CI: 1.09–1.35) and higher education (IRR:1.43, CI: 1.25–1.63), and women with a desire for pregnancy (IRR: 1.23, CI: 1.12–1.36) had a significantly higher frequency of ANC visits, while women from rural areas (IRR: 0.83, CI: 0.77–0.89) and women without decision-making power on their own healthcare issues (IRR: 0.87, CI: 0.78–0.97) had a significantly lower frequency of ANC visits. A lower (higher) frequency of ANC visits was almost invariably matched by a higher (lower) probability of taking no ANC at all during pregnancy. The estimated ORs shown in Fig 3 indicate that women from the Khulna (OR: 0.49, CI: 0.31–0.78) and Rangpur (OR: 0.62, CI: 0.40–0.96) divisions, from the middle (OR: 0.64, CI: 0.49–0.84), richer (OR: 0.42, CI: 0.30–0.58) and richest (OR: 0.22, CI: 0.13–0.35) households, women with primary (OR: 0.73, CI: 0.57–0.93), secondary (OR: 0.45, CI: 0.34–0.59) and higher education (OR: 0.24, CI: 0.14–0.44), women who have partners with secondary (OR: 0.62, CI: 0.48–0.81) and higher education (OR: 0.47, CI: 0.30–0.74) and those women who have access to mass media at least once a week (OR: 0.58, CI: 0.46–0.73) were significantly less likely to have no ANC visit.

The estimated variance components in the count part ($\sigma_c^2 = 0.069$) and the zero part ($\sigma_Z^2 = 0.626$) indicate significant community-level variation in the number of ANC visits, due to between-cluster heterogeneity.

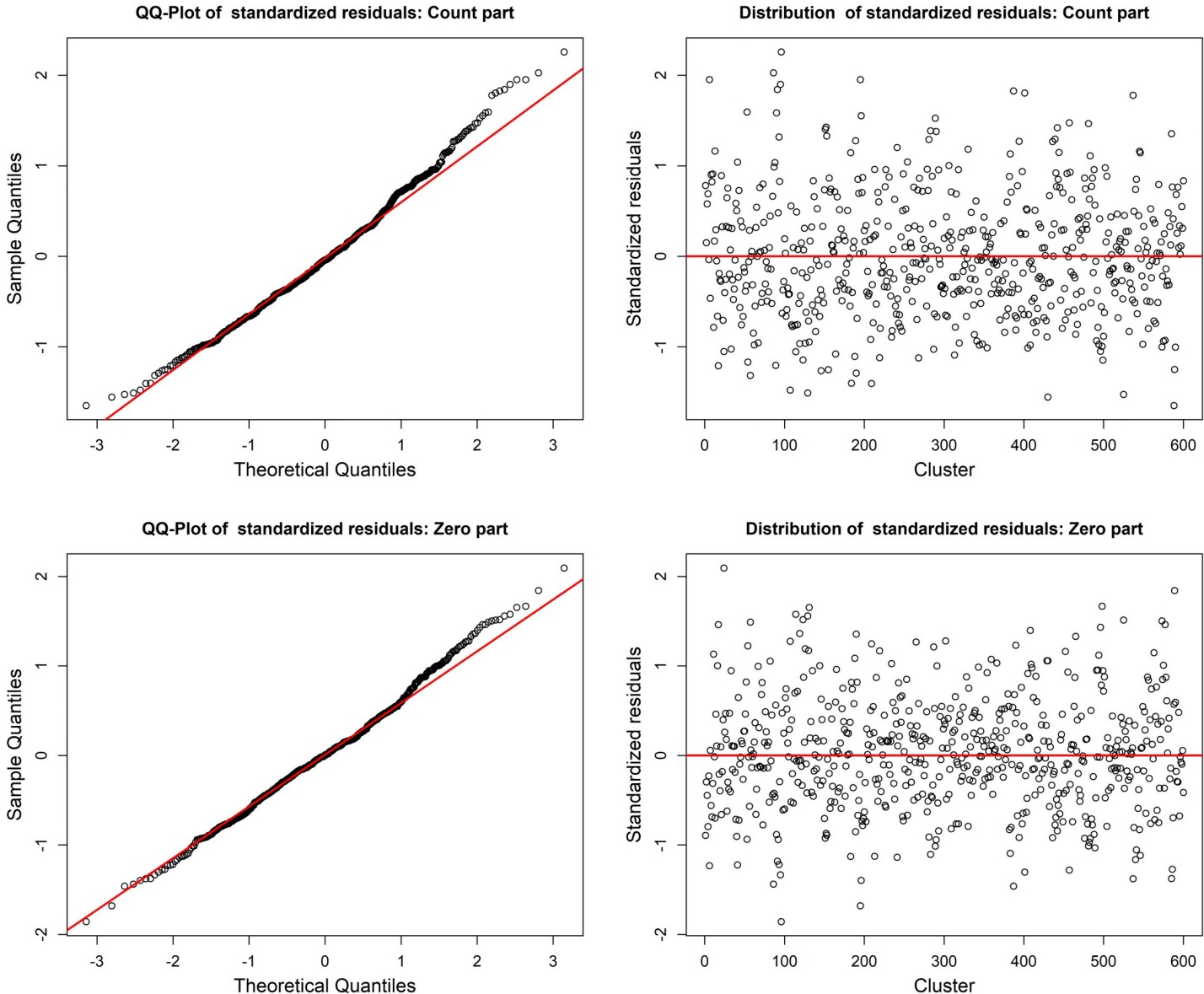

**Fig 2. Model diagnostics of both the count and the zero parts of the fitted mixed-effect hurdle negative binomial regression model through Q-Q and distribution plots of the standardized cluster-specific residuals.**

## Discussion

Aim of this study was to identify an appropriate count regression model for the number of ANC visits among pregnant women in Bangladesh utilizing recent nationally representative survey data. Since a substantial proportion of women did not take any prenatal care and the women are clustered according to the survey design, the performance of the standard Poisson and negative-binomial regression models have been compared with their zero-inflated and hurdle models with and without consideration of ICC in the model selection process.

The study has followed a systematic procedure to select the most appropriate count regression model for the frequency of ANC visits by examining a variety of criteria, particularly the existence of zero-inflation and community effects in the responses. It is found that the zero

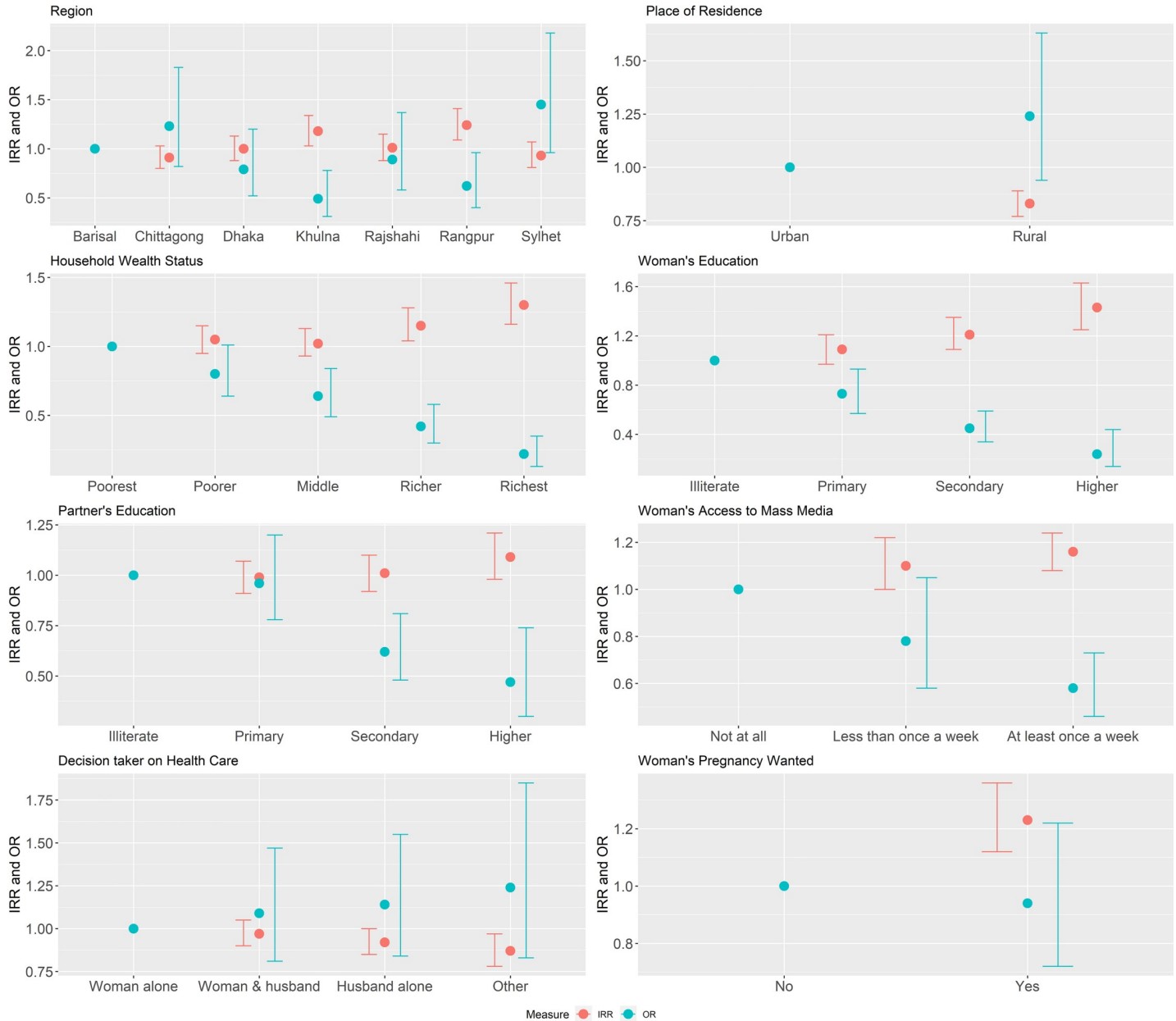

**Fig 3. Estimated incidence rate ratio (IRR) of having ANC visits (red dot and confidence line) and odds ratio (OR) of not attending any ANC visit (blue dot and confidence line) with 95% confidence interval (CI) from the hurdle negative binomial regression with random intercept at both count- and zero-part (MHNBR. ERE) models.**

ANC visits are generated from two different processes and hence either zero-inflated or hurdle regression model should be used to model the frequency of ANC visits in Bangladesh. Since the household surveys in Bangladesh use a complex cluster sampling design, regression models should also incorporate correlation (unless the considered explanatory variables in the model could explain the cluster-level variability), to prevent biased estimates with unfortunate under-coverage due to lower standard errors [32]. In this study, the incorporation of the ICC along with the help of uniformity tests facilitated the selection of the mixed-effect hurdle model as the appropriate model for the considered data.

**Table 6. Estimated regression coefficient (β), incidence rate ratio (IRR) of having ANC visits and odds ratio (OR) of not attending any ANC visit with their 95% CI and p-values from the hurdle negative binomial regression with random intercept at both count- and zero-part (MHNBR.ERE) models.**

| Factors | Category | Count-part (Number of ANC visits) | | | | Zero-part (No ANC attendance) | | | | |
|---|---|---|---|---|---|---|---|---|---|---|
| | | β | IRR | 95% CI | | p-value | β | OR | 95% CI | | p-value |
| Region | Barisal [Reference] | | | | | | | | | | |
| | Chittagong | -0.10 | 0.91 | 0.80 | 1.03 | 0.14 | 0.20 | 1.23 | 0.82 | 1.83 | 0.32 |
| | Dhaka | 0.00 | 1.00 | 0.88 | 1.13 | 0.96 | -0.24 | 0.79 | 0.52 | 1.20 | 0.27 |
| | Khulna | 0.17 | 1.18 | 1.03 | 1.34 | 0.01 | -0.71 | 0.49 | 0.31 | 0.78 | 0.00 |
| | Rajshahi | 0.01 | 1.01 | 0.88 | 1.15 | 0.91 | -0.12 | 0.89 | 0.58 | 1.37 | 0.59 |
| | Rangpur | 0.21 | 1.24 | 1.09 | 1.41 | 0.00 | -0.47 | 0.62 | 0.40 | 0.96 | 0.03 |
| | Sylhet | -0.07 | 0.93 | 0.81 | 1.07 | 0.33 | 0.37 | 1.45 | 0.96 | 2.18 | 0.08 |
| Place of Residence | Urban [Reference] | | | | | | | | | | |
| | Rural | -0.19 | 0.83 | 0.77 | 0.89 | 0.00 | 0.21 | 1.24 | 0.94 | 1.63 | 0.13 |
| Wealth Status | Poorest [Reference] | | | | | | | | | | |
| | Poorer | 0.05 | 1.05 | 0.95 | 1.15 | 0.35 | -0.22 | 0.80 | 0.64 | 1.01 | 0.07 |
| | Middle | 0.02 | 1.02 | 0.93 | 1.13 | 0.65 | -0.45 | 0.64 | 0.49 | 0.84 | 0.00 |
| | Richer | 0.14 | 1.15 | 1.04 | 1.28 | 0.01 | -0.87 | 0.42 | 0.30 | 0.58 | 0.00 |
| | Richest | 0.26 | 1.30 | 1.16 | 1.46 | 0.00 | -1.54 | 0.22 | 0.13 | 0.35 | 0.00 |
| Woman's Education | Illiterate [Reference] | | | | | | | | | | |
| | Primary | 0.08 | 1.09 | 0.97 | 1.21 | 0.14 | -0.32 | 0.73 | 0.57 | 0.93 | 0.01 |
| | Secondary | 0.19 | 1.21 | 1.09 | 1.35 | 0.00 | -0.81 | 0.45 | 0.34 | 0.59 | 0.00 |
| | Higher | 0.36 | 1.43 | 1.25 | 1.63 | 0.00 | -1.41 | 0.24 | 0.14 | 0.44 | 0.00 |
| Partner's Education | Illiterate [Reference] | | | | | | | | | | |
| | Primary | -0.02 | 0.99 | 0.91 | 1.07 | 0.73 | -0.04 | 0.96 | 0.78 | 1.20 | 0.73 |
| | Secondary | 0.01 | 1.01 | 0.92 | 1.10 | 0.89 | -0.47 | 0.62 | 0.48 | 0.81 | 0.00 |
| | Higher | 0.09 | 1.09 | 0.98 | 1.21 | 0.10 | -0.76 | 0.47 | 0.30 | 0.74 | 0.00 |
| Mass Media Exposure | Not at all [Reference] | | | | | | | | | | |
| | Less than once a week | 0.10 | 1.10 | 1.00 | 1.22 | 0.06 | -0.25 | 0.78 | 0.58 | 1.05 | 0.11 |
| | At least once a week | 0.15 | 1.16 | 1.08 | 1.24 | 0.00 | -0.55 | 0.58 | 0.46 | 0.73 | 0.00 |
| Pregnancy Wanted | No [Reference] | | | | | | | | | | |
| | Yes | 0.21 | 1.23 | 1.12 | 1.36 | 0.00 | -0.06 | 0.94 | 0.72 | 1.22 | 0.65 |
| Decision on health care | Woman alone [Reference] | | | | | | | | | | |
| | Woman & husband | -0.03 | 0.97 | 0.90 | 1.05 | 0.42 | 0.09 | 1.09 | 0.81 | 1.47 | 0.57 |
| | Husband alone | -0.08 | 0.92 | 0.85 | 1.00 | 0.05 | 0.13 | 1.14 | 0.84 | 1.55 | 0.41 |
| | Other | -0.14 | 0.87 | 0.78 | 0.97 | 0.01 | 0.22 | 1.24 | 0.83 | 1.85 | 0.29 |
| Intercept | | 0.66 | 1.93 | 1.61 | 2.32 | 0.00 | -0.14 | 0.87 | 0.51 | 1.48 | 0.61 |

Based on the selected hurdle regression model, women living in the Khulna and Rangpur divisions had a significantly lower probability of attending no antenatal care, compared to those living in the Sylhet and Chittagong divisions, while women from the Khulna and Rangpur divisions also had significantly higher frequency of ANC visits compared to women from the Sylhet and Chittagong divisions. These findings are highly supported by the findings obtained in a very similar study on ANC visits by Rahman et al. [33]. The findings may suggest that large-scale maternal and neonatal health programs worked properly in the economically poor Khulna and Rangpur regions compared to the economically rich Chittagong and Sylhet regions [34]. Another explanation could be worse access to maternal health services for the women who live in the remote hill-tract areas of the Chittagong division [35] and in the *haor* areas (a wetland ecosystem in the north-eastern part of Bangladesh) of the Sylhet division.

Guliani, Sepehri and Serieux [22] showed that women living in urban settings are more likely to attend prenatal care and have a higher frequency of visits compared to their counterparts, based on ANC data of 32 developing countries including Bangladesh. The results from the present study also indicate that women residing in urban areas have a higher frequency of ANC visits than those in rural areas. However, in our multivariate analysis, place of residence did not have a statistically significant influence on the woman's attitude to use ANC during pregnancy. The difference between frequency and use by urban-rural settings may arise mainly from the attitude of married adolescent women living in rural areas, who are less likely to use skilled maternal health services than those residing in urban areas [36–38]. The higher IRR for urban areas supports the idea that availability of health care centres has increased the access to maternal health services for urban women compared to the rural women. Moreover, women living in urban areas are relatively more educated, are more aware of health, and have more decision-making power on their own healthcare issues compared to women living in rural areas.

A positive association between the ANC utilization and the household wealth status has been found in many studies on ANC use [22, 33, 39]. This positive association does not vary by urban-rural settings [40]. The estimated OR and IRR in this study indicate that the probability to take no ANC and the frequency of ANC visits both increased with increasing household wealth status. A possible explanation could be that women who belong to well-off families usually have proper education, access to mass media, and an ability to spend more money to take frequent ANC visits compared to women from poorer families.

The findings of this study showed that women who have access to mass media at least once a week are less likely to keep away from ANC visits and have more ANC visits. Some studies on ANC utilization support this finding, particularly for women living in rural [39] and slum areas of big cities like Rajshahi [41] and Dhaka [42]. Mass media broadcast different sorts of health-related programs and news that make women aware of their well-being and the well-being of their unborn baby.

The likelihood of prenatal care attendance and the frequency of ANC use are both positively associated with the level of women's education and the influence of education is more pronounced for seeking prenatal care than the number of ANC visits [22]. The current study also found that the level of education had a stronger impact than other factors on both the use and frequency of ANC visits. Educated women took more ANC since they have more knowledge of the benefits of frequent ANC visits such as a reduction of pregnancy complications, ensuring safe delivery, and supporting healthy life of the babies. Moreover, they are more knowledgeable about how to find health care.

The findings of this study show that the partner's education also contributes to deciding whether a woman will take ANC, rather than the frequency of ANC visits. The probability of avoiding ANC significantly decreased with an increase of the partner's education status. Rahman, Islam and Islam [41] also found that the husband's education has a significant influence on taking prenatal care. The findings suggest that educated partners may be more concerned with their pregnant wives and the associated pregnancy complications.

The desire of pregnancy has a significant influence on the number of ANC visits in this study, rather than on the decision to seek ANC. Rahman et al. [33] also found that women are more likely to seek care for pregnancy complications when they intended to have the pregnancy. Conversely, when women are unwilling and unhappy about untimely pregnancy, they may be more likely to hide it and less likely to take frequent ANC visits. Hiding behaviour is common among women who live in a more conservative rural environment.

Women empowerment in health care decision-making is also found to be significantly associated with the number of ANC visits rather than seeking ANC. Women who can take

decisions by themselves take ANC visits more frequently than their counterparts do. A possible reason behind this finding could be that educated woman living in urban areas (who usually take decision by themselves) are more conscious about their own and their unborn babies health compared to illiterate women who depend on other's decision to seek prenatal care. Hossain and Hoque [11] also found a significant positive influence of women empowerment (measured by education, freedom of choice/movement, household decision making power, and economic activities) on the decision and intensity of utilization of antenatal care in Bangladesh.

## Conclusion

The selected hurdle regression model confirms that two processes generate the number of ANC visits in Bangladesh: one process generates zero ANC visits and the other generates the frequency of ANC visits. The significance of the cluster-specific variance component at both the zero and the count part of the hurdle regression model indicates that the community (cluster) has a significant effect on the variation of both the women's decision on prenatal care use and the frequency of ANC visits, although most of the variations originated from women-, household-, and regional- level factors. The findings of the study thus show the necessity of considering community effects (ICC) along with overdispersion and zero-inflation in modelling the ANC data of Bangladeshi women and hence also in identifying risk factors for not attending any ANC as well as for the frequency ANC visits. Though only random intercept models have been investigated in this study, further investigations can be performed to assess the relevance of random slopes in the model. Also, clustering at higher administrative units (such as district and sub-district) can be investigated using three- or four- level models [43]. Moreover, we found that hurdle and zero-inflated type models should be selected carefully since poorer assumptions of one type of (structural) zeros are difficult to derive from real world data. It is better to select the model structure statistically (whether the fitted model can explain all the zeros) rather than based on types of zeros.

The findings of this study might help policy makers to find out which socio-economic and demographic groups should be given priority to encourage women to attend ANC and to have more ANC visits to medically trained personnel during their pregnancy period. This study also suggests that besides improving women's academic education and household wealth, women should be motivated to change their attitude to seek medical care during their pregnancy. The significant cluster-level variation in the developed model also indicates that the goal of reducing maternal death could be achieved if heterogeneity in the prenatal care use and its frequency could be reduced at the community level.

## Supporting information

**S1 File. The minimal dataset extracted from 2014 BDHS for analysis.**
(CSV)

**S2 File. The R code used for analysis and creating graphs.**
(TXT)

## Acknowledgments

We thank DHS Macro Internationals for the permission to use the 2014 BDHS data set for this work. We acknowledge Dr. Dimitris Rizopoulos, Erasmus University Medical Center, The Netherlands for his assistance in performing analyses during the course of this research. We also thank Dr. Wilbert van den Hout and Dr. Ewout Steyerberg, Leiden University Medical

Center, The Netherlands, and Dr. M.R.A. Willems, Statistics Netherlands (CBS), Heerlen, the Netherlands for their valuable comments on the manuscript and suggestions in editing the final version of the article. Neither the original collectors of the data nor the Data Archive bears any responsibility for the analyses or interpretations presented in this article.

## Author Contributions

**Conceptualization:** Kakoli Rani Bhowmik, Sumonkanti Das.

**Data curation:** Kakoli Rani Bhowmik.

**Formal analysis:** Kakoli Rani Bhowmik, Md. Atiqul Islam.

**Investigation:** Sumonkanti Das.

**Methodology:** Kakoli Rani Bhowmik, Sumonkanti Das, Md. Atiqul Islam.

**Resources:** Kakoli Rani Bhowmik.

**Software:** Sumonkanti Das.

**Supervision:** Sumonkanti Das, Md. Atiqul Islam.

**Validation:** Kakoli Rani Bhowmik, Sumonkanti Das.

**Visualization:** Kakoli Rani Bhowmik, Sumonkanti Das, Md. Atiqul Islam.

**Writing – original draft:** Kakoli Rani Bhowmik, Sumonkanti Das, Md. Atiqul Islam.

**Writing – review & editing:** Kakoli Rani Bhowmik, Sumonkanti Das, Md. Atiqul Islam.

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
