## [Decision Letter · Decision Letter 0]

20 Nov 2019

PONE-D-19-25813

Modelling the Number of Antenatal Care Visits for Determining the Risk Factors of Antenatal Care Attendance and Its Frequency in Bangladesh

PLOS ONE

Dear Mr. Islam,

Thank you for submitting your manuscript to PLOS ONE. After careful consideration, we feel that it has merit but does not fully meet PLOS ONE’s publication criteria as it currently stands. Therefore, we invite you to submit a revised version of the manuscript that addresses the points raised during the review process.

We would appreciate receiving your revised manuscript by 19th December 2019. To enhance the reproducibility of your results, we recommend that if applicable you deposit your laboratory protocols in protocols.io, where a protocol can be assigned its own identifier (DOI) such that it can be cited independently in the future. For instructions see: http://journals.plos.org/plosone/s/submission-guidelines#loc-laboratory-protocols

We look forward to receiving your revised manuscript.

Kind regards,

Russell Kabir, PhD

Academic Editor

PLOS ONE

Journal Requirements:

2. Please indicate in your Data availability statement the link or contact information for the dataset used in this study (https://journals.plos.org/plosone/s/data-availability)

5. Please amend either the title on the online submission form (via Edit Submission) or the title in the manuscript so that they are identical.

Reviewers' comments:

Reviewer's Responses to Questions

**Comments to the Author**

1. Is the manuscript technically sound, and do the data support the conclusions?

Reviewer #1: Yes

Reviewer #2: Yes

Reviewer #3: Yes

2. Has the statistical analysis been performed appropriately and rigorously? 

Reviewer #1: Yes

Reviewer #2: Yes

Reviewer #3: Yes

3. Have the authors made all data underlying the findings in their manuscript fully available?

Reviewer #1: Yes

Reviewer #2: Yes

Reviewer #3: Yes

4. Is the manuscript presented in an intelligible fashion and written in standard English?

Reviewer #1: Yes

Reviewer #2: Yes

Reviewer #3: Yes

5. Review Comments to the Author

Reviewer #1: The authors propose a regression model and determine the risk factor of not using antenatal care during pregnancy. The system used several methodologies for finding the goals and they compared their proposed model with others. This is a very interesting work. However, few more clarification or issues may be addressed to improve the quality of the paper.

In the Statistical Models sub-section, the description of the proposed regression model should be more specific.

Author(s) highlight the risk factors explicitly in Abstract, Introduction, and DIscussion.

In Figure 1, the labels should be more specific to understand the chart properly

Reviewer #2: The study is essential for the improvement of the healthcare service in the country, especially the reproductive health. However, in findings and discussion, the study needs a bit more justified argument to support the claims. Also, few references are incomplete. It would be better if it can be completed for greater audience. The manuscript should also go through spell and grammar check before submission

Reviewer #3: This profound work includes some rigorous statistical modelling on antenatal care strategies which is quite different from the present-day works and conclusive in contest to data analysis. Though the outcomes may not be reasonable to the contemporary situations (as data were taken from 2014) but this work established some ground breaking results that will certainly reproduce by using updated data.

6. PLOS authors have the option to publish the peer review history of their article (what does this mean?). If published, this will include your full peer review and any attached files.

Reviewer #1: Yes: Sheikh Abujar

Reviewer #2: No

Reviewer #3: No

---

## [Author Response · Author response to Decision Letter 0]

1 Jan 2020

Comments by academic editor

Response: We have checked carefully the additional requirements and addressed as per instruction.

2. Please indicate in your Data availability statement the link or contact information for the dataset used in this study (https://journals.plos.org/plosone/s/data-availability)

Response. The statement has been still same. However, we added the DHS link from where the data can be obtained freely upon request.

Response: As per your suggestion, we requested two of our colleagues for reading and editing the article. They gave us nice corrections and feedbacks based on which we have improved the article remarkably in terms of language usage, spelling, and grammar. Based on their corrections and suggestion we have slightly modified the Title of the paper and also incorporated the Supplementary Table in the main text as a Table for convenience of the readers. Their corrections and modifications are revealed in the track change file. We have mentioned the logics behind the modification of the Title below in the response of Reviewer 2. 

The name of the colleague who edited our manuscript: 

Dr. Wilbert van den Hout, Leiden University Medical Center, Leiden, The Netherlands, E-mail: W.B.van_den_Hout@lumc.nl

Dr. M.R.A. Willems, Statistics Netherlands (CBS), Heerlen, the Netherlands, E-mail: rma.willems@cbs.nl

We have provided others information regarding the aforementioned three bullets in the online submission system.

Comments by reviewers

Reviewer #1: The authors propose a regression model and determine the risk factor of not using antenatal care during pregnancy. The system used several methodologies for finding the goals and they compared their proposed model with others. This is a very interesting work. However, few more clarification or issues may be addressed to improve the quality of the paper.

In the Statistical Models sub-section, the description of the proposed regression model should be more specific.

Response: Many thanks for the suggestion. The Poisson and Negative Binomial regression models are well known and so these models are not described elaborately. Instead, Zero-inflated and Hurdle regression models are illustrated more in simplest way. After rereading the paper, we find there are some problems in denoting the models. Now we adjust these problems by providing clear sentences for the specific types of model as below:

“The mixed-effects PR and NBR models are referred to hereafter as MPR and MNBR models respectively.”

“According to the modelling distribution (Poisson or negative binomial) of the second so-called count part of the process, the ZIR and HR models are referred to as either ZIPR/HPR or ZINBR/HNBR.”

“When cluster-specific random effects are considered only in the count part (so c_0j=0), the mixed-effects ZIR and HR models are denoted by MZIPR/MZINBR and MHPR/MHNBR respectively depending on the assumed distribution (Poisson or negative binomial) of the count part. The mixed models with extra random effects at the zero component Models with also a random effect in the zero part (usually it is considered as an extra random effect in the mixed-effects ZIR/HR model) are denoted hereafter as MZINBR.ERE (for example) for the MZINBR model.”

Author(s) highlight the risk factors explicitly in Abstract, Introduction, and Discussion.

Response: The second goal of the study is to find determinants of taking ANC and the frequency of ANC visits. So, we discussed about determinants in Abstract, Introduction and Discussion. However, we focused more on modelling issues as the first objective is to find a suitable model for the number of ANC visits. We discuss in detail about determinants only in the Discussion section. We compared the findings of this study with other studies and illustrate some arguments regarding the determinants obtained from the fitted model. As per the comments of third reviewer, we have slightly modified the discussion section. 

In Figure 1, the labels are more specified to understand the chart properly. 

Response: We have increased the font side of the labels of the figure to be more specific. Also, the graph is prepared following the PLOS ONE suggestion.

Reviewer #2: 

The study is essential for the improvement of the healthcare service in the country, especially the reproductive health. However, in findings and discussion, the study needs a bit more justified argument to support the claims. 

Response: We have tried to provide our claims with reference where possible. Some discussions are based on the study findings and so these are not possible to justified with other references. We have revised the last paragraph of the Discussion section with a relevant reference to justify our claims as below:

“Hossain and Hoque [11] also found a significant positive influence of women empowerment (measured by education, freedom of choice/movement, household decision making power, and economic activities) on the decision and intensity of utilization of antenatal care in Bangladesh.” 

Also, few references are incomplete. It would be better if it can be completed for greater audience. 

Response: Incomplete references have been completed and provided with the manuscript. The file with track change shows the detail change in the references with complete information. In addition, we checked all references as the reference format of the paper. 

The manuscript should also go through spell and grammar check before submission

Response: The manuscript has been read by two colleagues of two authors. They made extensive correction and give nice feedback. As their suggestions, we have made the corrections and changes some sentences to make them standard and simpler. Based on the readers corrections, the writing style of the paper has been improved remarkably now. Based on the suggestion of one colleague, we have changed our Title of the article. The current title exactly reflects the aim of the study. The new title is now “Modelling the Number of Antenatal Care Visits in Bangladesh to Determine the Risk Factors for Reduced Antenatal Care Attendance” instead of “Modelling the Number of Antenatal Care Visits for Determining the Risk Factors of Antenatal Care Attendance and Its Frequency in Bangladesh”. The logics behind the title is as below according to the reader: 

1. I would replace the first “frequency” by “number”, because both use (0 or not) and frequency (count number) are modelled

2. I added “reduced” because risk is usually associated with negative things (and attendance is positive)

3. I removed “and its frequency” at the end because it is shorter and most readers will not understand that you mean to say already that taking ANC and the frequency of ANC may be different processes.

We agreed with his logics and so, we modified the title which is now simpler than before. The first objective is to model the number of ANC visits and the second objective is to find determinants of reduced number of ANC visits (which cover both zero and lower frequency of ANC visits).

 

Reviewer #3:

This profound work includes some rigorous statistical modelling on antenatal care strategies which is quite different from the present-day works and conclusive in contest to data analysis. 

Though the outcomes may not be reasonable to the contemporary situations (as data were taken from 2014) but this work established some ground breaking results that will certainly reproduce by using updated data.

Response: Many thanks for the comments on analysis. The dataset used in this study is the only available updated DHS dataset available for Bangladesh. The most recent survey has been conducted in 2017-2018. Though preliminary report has just been published (December 2019), the full report and data are not yet available. After getting the new data, the similar analysis can be easily conducted, and we hope new findings will be found based on the new data.

---

## [Editor Report · Decision Letter 1]

10 Jan 2020

Modelling the Number of Antenatal Care Visits in Bangladesh to Determine the Risk Factors for Reduced Antenatal Care Attendance

PONE-D-19-25813R1

Dear Dr. Islam,

We are pleased to inform you that your manuscript has been judged scientifically suitable for publication and will be formally accepted for publication once it complies with all outstanding technical requirements.

With kind regards,

Russell Kabir, PhD

Academic Editor

PLOS ONE
---

## [Editor Report · Acceptance letter]

15 Jan 2020

PONE-D-19-25813R1 

Modelling the Number of Antenatal Care Visits in Bangladesh to Determine the Risk Factors for Reduced Antenatal Care Attendance 

Dear Dr. Islam:

I am pleased to inform you that your manuscript has been deemed suitable for publication in PLOS ONE. Congratulations! Your manuscript is now with our production department. 

With kind regards,

on behalf of

Dr. Russell Kabir 

Academic Editor

PLOS ONE